

# Evaluation of the socially evaluated cold-pressor group test (SECPT-G) in the general population

Linda Becker, Ursula Schade and Nicolas Rohleder

Institute of Psychology, Friedrich-Alexander-Universität Erlangen-Nürnberg, Erlangen, Germany

## ABSTRACT

**Background:** In stress research, economic instruments for introducing acute stress responses are needed. In this study, we investigated whether the socially evaluated cold-pressor group test (SECPT-G) induces salivary alpha-amylase (sAA) and/or cortisol responses in the general population and whether this is associated with anthropometric, experimental, and lifestyle factors.

**Methods:** A sample of 91 participants from the general population was recruited. Salivary cortisol and sAA levels were assessed prior to ($t_0$), immediately after ($t_1$), and 10 min after the SECPT-G ($t_2$).

**Results:** A robust cortisol increase was found immediately after the SECPT-G, which further increased between $t_1$ and $t_2$. This was independent of most of the control variables. However, men showed a trend toward higher cortisol increases than women ($p = 0.005$). No sAA responses were found at all. However, sAA levels were dependent on measurement time point with highest levels between 9 pm and 9:30 pm. Participants who immersed their hands into the ice water for the maximally allowed time of 3 min showed higher sAA levels at all time points than participants who removed their hands from the water earlier.

**Conclusions:** We conclude that the SECPT-G is a good means of an acute stress test when cortisol—but not necessarily sAA—responses are intended.

Corresponding author
Linda Becker, linda.becker@fau.de

## INTRODUCTION

Stress is associated with a variety of physiological, emotional, and cognitive processes as well as with several disorders (e.g., cardiovascular diseases and depression). However, the processes underlying the acute stress response have not yet been fully understood. Therefore, experimental set-ups are needed that allow eliciting stress responses in the laboratory.

One standard procedure in stress research is the socially evaluated cold-pressor test (SECPT; *Schwabe, Haddad & Schachinger, 2008*). The SECPT combines a physiological stressor (immersing one's hand in ice water; *Lovallo, 1975*) with socially-evaluative components (being watched by the experimenter and being videotaped by a camera). The SECPT is an economic alternative to public speaking paradigms (e.g., the trier social stress

test (TSST); *Kirschbaum, Pirke & Hellhammer, 1993*) which are labor intensive and, therefore, an impediment for recruiting larger samples. *Minkley et al. (2014)* showed that the SECPT can also be performed in groups (socially evaluated cold-pressor test for groups, SECPT-G) and that this is, thus, an even more economic variant of the original SECPT set-up. Minkley and colleagues evaluated the SECPT-G in a sample of 61 middle-aged, normal weight, non-smoking participants. They found significant cardiovascular (i.e., an increase in blood pressure and heart rate) and hypothalamic-pituitary-adrenal (HPA) axis responses (i.e., an increase in cortisol levels).

However, it has previously been shown that the acute stress response is associated with a variety of demographic, anthropometric, and lifestyle factors. In particular, associations with participants' sex have been repeatedly reported. For example, stronger HPA axis responses in young men than in young women have been found (*Kirschbaum, Wüst & Hellhammer, 1992*; *Kudielka & Kirschbaum, 2005*; *Stephens et al., 2016*). Furthermore, a delayed post stress recovery has been found in women (*Owen et al., 2003*). Moreover, in women, the stress response has been associated with the phase of the menstrual cycle and with the use of oral contraceptives (*Kirschbaum et al., 1999*). Regarding age, it has been found that it is negatively related with HPA axis response (i.e., the cortisol secretion after an acute stressor is decreased in older adults; *Kudielka et al., 2004*).

The findings with regard to body composition (i.e., with the body mass index (BMI)) are divergent in that some authors reported positive and some others reported negative associations between BMI and the cortisol response to acute stressors (*Jones et al., 2012*; *McInnis et al., 2014*). Furthermore, the stress response is associated with socio-economic factors. For example, *Owen et al. (2003)* found stronger HPA axis responses in people with low socio-economic statuses and low incomes. These factors are related with chronic stress, which is associated with the stress response as well (*Kudielka, Bellingrath & Hellhammer, 2006*).

Lifestyle factors can also influence HPA axis reactivity. One of the best studied factors is smoking which leads to chronically elevated cortisol levels and to reduced responses to acute stressors (*Kirschbaum, Wüst & Strasburger, 1992*; *Kudielka, Hellhammer & Wüst, 2009*; *Rohleder & Kirschbaum, 2006*). Furthermore, caffeine consumption can affect the acute stress response, leading to higher salivary alpha-amylase (sAA), cortisol, and cardiovascular reactivity (*Klein et al., 2010*; *Lane et al., 1990*; *Lane & Williams, 1985*). A further very important lifestyle factor is regular physical activity which affects HPA axis activity and, therefore, the response to acute stressors (*Luger et al., 1987*).

The results of the studies summarized above underscore that it is necessary that an evaluation of a stress paradigm should be performed in a broad population and that the associations with demographic, anthropometric, and lifestyle factors should be considered. Therefore, in our study, we recruited a sample from the general population (i.e., including almost all age and weight groups, as well as smokers). Besides, in none of the previous studies it has been investigated whether the SECPT-G also leads to an increase in sAA secretion. Some authors suggest—although there are some valid concerns that need to be taken into account (*Bosch et al., 2011*)—that sAA can be used as a marker for sympathetic nervous system activity (*Nater et al., 2007*; *Rohleder & Nater, 2009*) and, thus, it should be

investigated in stress studies as well. Therefore, in our study, we investigated whether—beside a cortisol increase—an sAA response could be elicited by means of the SECPT-G as well. Our approach was threefold. First, we investigated whether the SECPT-G introduces an HPA axis response (i.e., a cortisol increase). Second, we examined whether the SECPT-G also induces a SNS response (i.e., an increase in sAA). Third, we explored whether the stress response is associated with a variety of demographic, anthropometric, and lifestyle factors (e.g., age, BMI, sex, use of oral contraceptives, physical activity, smoking, chronic stress) as well as with experimental factors (e.g., time of day).

# MATERIALS AND METHODS

## Participants

The sample size was 96 ($N = 96$). The participants came to our laboratory in the context of a public event (open day of the university) and were then asked whether they would like to participate in a stress experiment. Because of missing data, five participants had to be excluded from statistical analysis. The remaining 91 participants ($N = 91$) had a mean age of $36.8 \pm 14.3$ years (min: 18 years, max: 73 years) and a BMI of $24.1 \pm 3.7$ kg/m$^2$ (min: 16.1, max: 35.4). All participants reported that they had not eaten or consumed beverages at least 1 h before the start of the experiment. Most of the participants were German ($N = 80$, 87.9%). A more detailed sample description is provided in the results section. All participants gave their written and informed consent. The study was carried out in accordance with the Code of Ethics of the World Medical Association (Declaration of Helsinki) and was approved by the local ethics committee of the Friedrich-Alexander University Erlangen-Nürnberg (# 6_18 B).

## Procedure

### General procedure

The experiment was performed 12 times in groups of eight participants on one evening between 06:30 pm and midnight. Each session lasted about 25 min. Participants were informed that they would take part in a stress experiment. After they gave their consent for participation, they waited in a room that was not the experimental room, where they disinfected their hands, and rinsed their mouth with water. This lasted about 5 min. After this, they were brought as a group to the experimental room where they were made familiar with the saliva collection procedure. Saliva was collected by means of salivettes (Sarstedt, Nümbrecht, Germany). During saliva collection, subjective stress perception was rated on a 10-point Likert scale with the anchors "not stressed at all" and "extremely stressed." Subsequently after instruction, the first saliva sample ($t_0$) was collected. After this, the SECPT-G (see below for further specifications) was explained and then started immediately. The second saliva sample ($t_1$) was collected immediately after the SECPT-G. To fill the gap between the third saliva sample ($t_2$) which was collected 10 min after the SECPT-G, participants filled out some questionnaires (see below).

### Stress induction

In the experimental room, all participants were asked to stand around a large table with transparent boxes filled with ice water in front of them. They were instructed to immerse

their hands in the ice water as long as possible for up to 3 min. Mean immersion time was 2:30 ± 0:55 min (max: 3:00 min, min: 0:39 min). The hand of each participant was directly opposite of the hand of another person with the aim to introduce a competitive situation. Remaining time was displayed on a large-display digital clock that was visible for all of the participants. An auditory countdown announced the last 5 s. Therefore, our protocol slightly differed from that reported by *Schwabe, Haddad & Schachinger (2008)* and *Minkley et al. (2014)* because in those previous studies no countdown was used. Another difference was that we did not use a camera. Two experimenters were present during the SECPT-G. They wore medical uniforms and were instructed to behave distanced and have a neutral mimic.

### *Assessment of demographic variables and lifestyle factors*

Between $t_1$ and $t_2$, participants filled out questionnaires which assessed demographic variables (e.g., age, sex, graduation, profession) as well as further control variables. Furthermore, participants were asked whether they were smokers, whether they were regularly consuming caffeine-containing beverages, and whether they had already consumed alcoholic beverages that evening. Participants that reported that they had consumed more than the equivalent of two alcohol-containing drinks or had consumed alcoholic beverages within 2 h before the experiment were screened out prior to the experiment. BMI was assessed via self-reports as well. We tried to keep this situation as pleasant as possible to avoid inducing a further stress response that might have masked the response to the SECPT-G. Chronic stress was measured by means of the 12-item screening scale of the trier inventory of chronic stress (TICS-SSCS; *Schulz & Schlotz, 1999*). This scale has been evaluated in a German sample and shows high internal consistency (Cronbach's $\alpha$ = 0.91; *Petrowski et al., 2012*). The amount of regular physical activity was measured by means of the short form of the international physical activity questionnaire (IPAQ; *Craig et al., 2003*). The IPAQ is a standard tool for assessing activity levels via self-reports and has been evaluated in different nationalities as well as age groups. The inter-reliability between the short and the long-form is 0.67 (*Craig et al., 2003*).

## Sample processing

Saliva samples were stored at −30 °C after collection for later analyses. Before cortisol and sAA measurement, two freeze-thaw cycles were performed. Immediately before measurement, samples were centrifuged at 2,000×$g$ and 20 °C for 10 min. sAA was measured with an in-house enzyme kinetic assay using reagents from Roche Diagnostics (Mannheim, Germany), as previously described (*Bosch et al., 2003*; *Rohleder & Nater, 2009*). In brief, saliva was diluted at 1:625 with ultrapure water, and diluted saliva was incubated with substrate reagent ($\alpha$-amylase EPS Sys; Roche Diagnostics, Mannheim, Germany) at 37 °C for 3 min before a first absorbance reading was taken at 405 nm with a Tecan Infinite 200 PRO reader (Tecan, Crailsheim, Germany). A second reading was taken after 5 min incubation at 37 °C and increase in absorbance was transformed to sAA concentration (U/ml), using a standard curve prepared using "Calibrator f.a.s." solution (Roche Diagnostics, Mannheim, Germany). Salivary cortisol concentrations were

determined in duplicate using chemiluminescence immunoassay (IBL, Hamburg, Germany). Intra- and inter-assay coefficients of variation were below 10% for both sAA and cortisol.

## Statistical analysis

For statistical analyses, IBM SPSS Statistics (version 26) was used. All control variables were categorized prior to statistical analysis. Age was grouped by means of a median split into two groups of younger (≤33 years) and older (>33 years) participants. BMI was classified according to the norms provided by the World Health Organization as underweight ($<18.5$ kg/m$^2$), normal weight (18.5–24.9 kg/m$^2$), pre-obese (25–29.9 kg/m$^2$), and obese ($>29.9$ kg/m$^2$). The amount of regular physical activity was categorized into low, moderate, and high amounts of regular physical activity (*Rangul et al., 2008*). Chronic stress levels were grouped into low- vs. high chronic stress groups by using the means that were provided by *Petrowski et al. (2012*, i.e., 12.9 for men and 13.7 for women*)*. Furthermore, participants were grouped into "winners" and "losers" according to their performance in the SECPT-G, that is, participants who put their hand in the ice water for the maximum time of 3 min were classified as "winners" and participants who put their hands out of the water earlier were classified as "losers." Normality of distribution was tested by means of the Kolmogorov–Smirnov test for the metric variables. Because of positive skewness and violation of normality, sAA and cortisol levels were transformed by means of the natural logarithm prior to further statistical analysis. Analyses of variance for repeated measurements (rmANOVAs) with the within-subject factor time ($t_0$, $t_1$, $t_2$) were calculated, separately for subjective stress ratings, sAA and cortisol levels. Partial eta-squares ($\eta_p^2$) were considered as effect sizes. Sphericity was tested by means of the Mauchly test (*Mauchly, 1940*). If necessary, degrees of freedom were corrected by means of the Greenhouse-Geisser procedure (*Greenhouse & Geisser, 1959*). For post hoc analysis, *t*-tests for dependent samples were calculated and Cohen's *d* was considered as measure for effect sizes. For these dependent *t*-tests, Cohen's *d* was corrected according to the method that was proposed by *Morris (2008)*.

To investigate whether one of the control variables was responsible for the main effect of the factor time, these variables were entered as additional factors into further rmANOVAs. For these analyses, adjusted alpha levels of $\alpha = 0.05/13 = 0.0038$ were used because 13 control variables (age, sex, BMI, smoking, caffeine, alcohol, use of oral contraceptives, education, profession, chronic stress, regular physical activity, time of day, winner-loser) were considered. For further analysis of significant effects of the control variables, *t*-tests for independent samples were calculated. Cohen's *d* was considered as measure for effect sizes. When reporting descriptive statistics in the text, mean ± standard deviations are provided. In Figs. 1–3, standard errors are used as error bars.

## Power analysis

Before the start of the experiment an a-priori power analysis was conducted by using GPower (version 3.1). We calculated the optimal sample size for a repeated measures ANOVA with within-between interaction for an $\alpha$-level of 0.05, a power of $1\text{-}\beta = 0.95$,

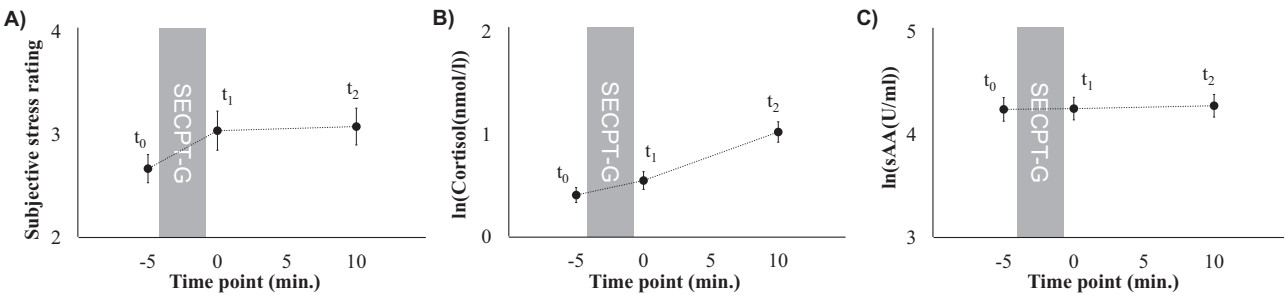

**Figure 1 Subjective stress ratings (A), mean cortisol levels (B), and mean sAA levels (C) prior to the SECPT-G ($t_0$), immediately after ($t_1$), and 10 min after it ($t_2$).**

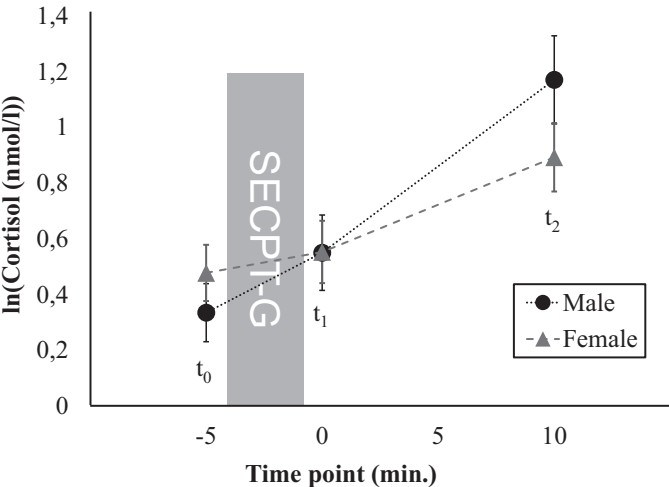

**Figure 2 Time course of the cortisol response, separately for men and women.**

12 groups (because the factor time of day had 12 values), and three measurement time points. This yielded an optimal sample size of $N = 96$. Unfortunately, we had to exclude five participants from statistical analysis. However, the effect sizes that we found and that are provided in the following section are much higher than the medium effect size that was entered into power analysis.

## RESULTS

### Descriptive statistics

A total of 43 of the participants were male, 11 were smokers, and 22 had already consumed alcoholic beverages on the experimental day, but no one had consumed more than the equivalent of two drinks and no one had consumed alcoholic beverages within 2 h before the experiment. Most of the participants were German. A total of 28 of the participants reported regular caffeine consumption. Seven of the female participants reported use of oral contraceptives. Mean activity levels were 5,216 ± 5,719 (min: 240, max: 28,770) metabolic minutes per week which refers to 6,229 ± 7,061 (min: 200, max: 31,647)

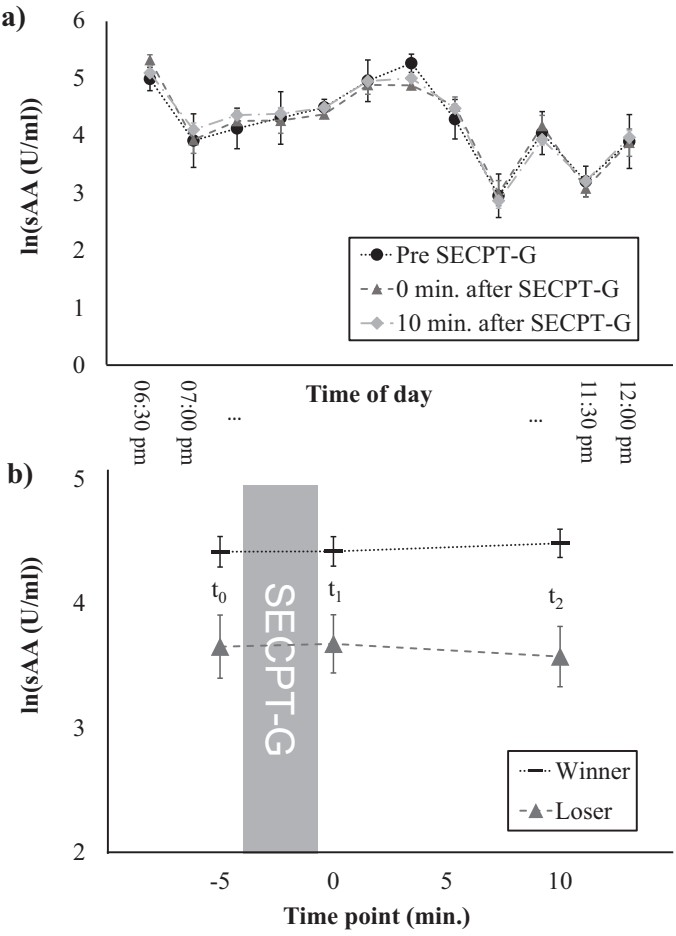

**Figure 3 Salivary α-amylase levels at different times of the day (A) and time course of the sAA response, separately for winners and losers (B).** Full-size  DOI: 10.7717/peerj.7521/fig-3

metabolic equivalents per week. Mean scores in the TICS-SSCS were 18.5 ± 7.12 (min: 4, max: 36). A detailed descriptive sample description is provided in Table 1.

## Subjective stress perception

Subjective stress perception that was rated on a 10-point Likert scale did not significantly differ between the three time points ($F(2, 180) = 2.75$, $p = 0.067$, $\eta_p^2 = 0.03$; $t_0$: 2.7 ± 1.3, $t_1$: 3.0 ± 1.8, $t_2$: 3.1 ± 1.7). However, there was a trend toward higher ratings after the SECPT-G ($t_0$ and $t_1$: $t(90) = -1.89$, $p = 0.062$, $d = 0.24$, $t_0$ and $t_2$: $t(90) = -2.22$, $p = 0.029$, $d = 0.28$, Fig. 1A).

## HPA axis response

Cortisol levels significantly increased after the SECPT-G ($F(1.56, 140.37) = 52.53$, $p < 0.001$, $\eta_p^2 = 0.37$; $t_0$: 0.41 ± 0.69, $t_1$: 0.55 ± 0.83, $t_2$: 1.02 ± 0.95 ln(nmol/l) or rather $t_0$: 1.9 ± 1.7, $t_1$: 2.6 ± 3.1, $t_2$: 4.3 ± 4.7 nmol/l; Fig. 1B). This effect was significant between all of the three time points ($t_0$ and $t_1$: $t(90) = -2.93$, $p = 0.004$, $d = 0.55$; $t_1$ and $t_2$: $t(90) = -4.62$, $p < 0.001$, $d = 0.44$; $t_0$ and $t_2$: $t(90) = -6.00$, $p < 0.001$, $d = 0.82$).

**Table 1 Descriptive statistics for all control variables that were entered in the statistical analysis.**

| Variable | Value | Frequency | Percentage |
|---|---|---|---|
| Sex | Male | 43 | 47.3 |
| | Female | 48 | 52.7 |
| Age | ≤33 years | 48 | 52.7 |
| | >33 years | 43 | 47.3 |
| BMI | Underweight | 5 | 5.5 |
| | Normal weight | 55 | 60.4 |
| | Pre-obese | 24 | 26.4 |
| | Obese | 7 | 7.7 |
| Education | Certificate of secondary education ("Hauptschulabschluss") | 1 | 1.1 |
| | Secondary school level ("Mittlere Reife") | 19 | 20.9 |
| | Graduation ("Ausbildung") | 5 | 5.5 |
| | Vocational diploma ("Fachabitur") | 10 | 11.0 |
| | General qualification for university entrance ("Abitur") | 23 | 25.3 |
| | Bachelor degree | 9 | 9.9 |
| | Diploma or master degree | 17 | 18.7 |
| | PhD | 4 | 4.4 |
| | Other | 2 | 2.2 |
| | Missing | 1 | 1.1 |
| Profession | Student | 17 | 18.7 |
| | Full-time employee | 35 | 38.5 |
| | Graduation | 1 | 1.1 |
| | Part-time employee | 14 | 15.4 |
| | PhD student | 4 | 4.4 |
| | Retired | 3 | 3.3 |
| | Self-employed | 8 | 8.8 |
| | Unemployed | 8 | 8.8 |
| | Missing | 1 | 1.1 |
| Smoking | No | 11 | 12.1 |
| | Yes | 80 | 87.9 |
| Caffeine | No | 14 | 15.4 |
| | Yes | 28 | 30.8 |
| | Missing | 49 | 53.8 |
| Alcohol | No | 61 | 67.0 |
| | Yes | 22 | 24.2 |
| | Missing | 8 | 8.8 |
| Oral contraceptives (women only) | No | 41 | 85.4 |
| | Yes | 7 | 14.6 |
| Activity level | Low | 17 | 18.7 |
| | Moderate | 24 | 26.4 |
| | High | 48 | 52.7 |
| | Missing | 2 | 2.2 |

| Table 1 (continued). | | | |
|---|---|---|---|
| **Variable** | **Value** | **Frequency** | **Percentage** |
| Chronic stress | Low stress | 47 | 51.6 |
| | High stress | 44 | 48.4 |
| SECPT performance | Winner | 71 | 78.0 |
| | Loser | 20 | 22.0 |

**Note:**
Sample size was $N = 91$.

## Alpha-amylase response

Mean sAA levels did not differ between the three time points ($F(2, 180) = 0.22$, $p = 0.801$, $\eta_p^2 = 0.002$; $t_0$: $4.2 \pm 1.1$, $t_1$: $4.3 \pm 1.0$, $t_2$: $4.3 \pm 1.0$ ln(U/ml) or rather $t_0$: $113.7 \pm 103.2$, $t_1$: $112.1 \pm 102.2$, $t_2$: $114.8 \pm 103.5$ U/ml, Fig. 1C).

## Associations with control variables

Summaries of the analyses of the control variables are provided in Table 2 for cortisol and in Table 3 for sAA.

### Anthropometric and demographic factors

First, we investigated whether sex, age, BMI, education, or profession were associated with the stress response. Therefore, these variables were included as additional factors in further rmANOVAs, separately for cortisol and sAA. For cortisol, main effects of time (i.e., increases in cortisol levels between $t_0$ and $t_2$) were found for all control variables (Table 2). After Greenhouse-Geisser correction, a marginally significant interaction time × sex was found ($F(1.59, 1) = 6.15$, $p = 0.005$, $\eta_p^2 = 0.07$; Fig. 2). Further post hoc ANOVAs yielded main effects of time for both men ($F(1.68, 70.60) = 31.72$, $p < 0.001$, $\eta_p^2 = 0.43$) and for women ($F(1.22, 57.53) = 25.37$, $p < 0.001$, $\eta_p^2 = 0.35$). No significant differences in mean cortisol levels were found between men and women for none of the three time points ($t_0$: $t(89) = -0.98$, $p = 0.328$, $d = 0.20$, $t_1$: $t(89) = -0.01$, $p = 0.989$, $d = 0.003$, $t_2$: $t(89) = 1.41$, $p = 0.164$, $d = -0.26$). For none of the other control variables interaction effects were found for cortisol. For none of the control variables, main effects were found for cortisol. For sAA, no main effects of time, no interactions time × control variable as well as no main effects of the control variables were found (Table 3).

### Lifestyle factors

Furthermore, we investigated whether the lifestyle factors smoking, caffeine or alcohol consumption, the use of oral contraceptives, the amount of regular physical activity or the perception of chronic stress were associated with the stress response. For cortisol, main effects of time (i.e., increases in cortisol levels between $t_0$ and $t_2$) were found for all lifestyle factors (Table 2). For none of the control variables, interaction effects with the factor time nor main effects were found for cortisol. For none of the control variables, main effects were found for cortisol. For sAA, no main effects of time, no interactions time × control variable as well as no main effects of the control variables were found (Table 3).

**Table 2 Associations between the cortisol time course and anthropometric, demographic, lifestyle, and experimental control factors.**

| Control variable | Main effect time | | Interaction time × control variable | | Main effect control variable | |
|---|---|---|---|---|---|---|
| | $p$ | $\eta_P^2$ | $p$ | $\eta_P^2$ | $p$ | $\eta_P^2$ |
| Sex | <0.001 | 0.39 | 0.005 | 0.07 | 0.781 | 0.001 |
| Age | <0.001 | 0.39 | 0.014 | 0.05 | 0.166 | 0.02 |
| BMI | 0.00001 | 0.17 | 0.328 | 0.04 | 0.638 | 0.02 |
| Education | 0.00002 | 0.14 | 0.050 | 0.15 | 0.141 | 0.14 |
| Profession | <0.001 | 0.21 | 0.137 | 0.11 | 0.332 | 0.09 |
| Smoking | 0.0002 | 0.11 | 0.019 | 0.05 | 0.061 | 0.04 |
| Caffeine | 0.000002 | 0.28 | 0.769 | 0.01 | 0.062 | 0.08 |
| Alcohol | <0.001 | 0.34 | 0.687 | 0.004 | 0.380 | 0.01 |
| Oral contraceptives | 0.00006 | 0.26 | 0.462 | 0.01 | 0.491 | 0.01 |
| Physical activity | <0.001 | 0.35 | 0.715 | 0.01 | 0.815 | 0.01 |
| Chronic stress | <0.001 | 0.37 | 0.324 | 0.01 | 0.660 | 0.002 |
| Time of day | 0.005 | 0.42 | 0.064 | 0.19 | 0.38 | 0.48 |
| SECPT-G performance | <0.001 | 0.30 | 0.792 | 0.002 | 0.865 | 0.0003 |

**Table 3 Associations between the time course of the sAA response and anthropometric, demographic, lifestyle, and experimental control factors.**

| Control variable | Main effect time | | Interaction time × control variable | | Main effect control variable | |
|---|---|---|---|---|---|---|
| | $p$ | $\eta_P^2$ | $p$ | $\eta_P^2$ | $p$ | $\eta_P^2$ |
| Sex | 0.795 | 0.003 | 0.759 | 0.003 | 0.692 | 0.002 |
| Age | 0.808 | 0.002 | 0.891 | 0.001 | 0.747 | 0.001 |
| BMI | 0.557 | 0.01 | 0.786 | 0.02 | 0.429 | 0.03 |
| Education | 0.504 | 0.01 | 0.087 | 0.13 | 0.477 | 0.09 |
| Profession | 0.247 | 0.02 | 0.715 | 0.06 | 0.845 | 0.04 |
| Smoking | 0.439 | 0.01 | 0.448 | 0.01 | 0.613 | 0.003 |
| Caffeine | 0.379 | 0.02 | 0.175 | 0.04 | 0.137 | 0.06 |
| Alcohol | 0.714 | 0.004 | 0.428 | 0.01 | 0.133 | 0.03 |
| Oral contraceptives | 0.433 | 0.02 | 0.206 | 0.03 | 0.440 | 0.01 |
| Physical activity | 0.981 | 0.0002 | 0.595 | 0.02 | 0.582 | 0.01 |
| Chronic stress | 0.790 | 0.003 | 0.740 | 0.003 | 0.455 | 0.006 |
| Time of day | 0.829 | 0.002 | 0.847 | 0.087 | 0.00001 | 0.41 |
| SECPT-G performance | 0.960 | 0.0005 | 0.402 | 0.01 | 0.001 | 0.11 |

*Experimental factors*

Finally, we investigated whether the time of day and the SECPT-G performance (i.e., being classified as a winner or loser) were associated with the cortisol and/or sAA response. For cortisol, main effects of time (i.e., increases in cortisol levels between $t_0$ and $t_2$) were found for both control variables (Table 2). For none of the experimental factors, interaction effects with the factor time nor main effects were found for cortisol. For sAA, neither main effects of time nor interactions time × control variable were found. However, main effects of time of day ($F(11, 78) = 24.87$, $p < 0.001$, $\eta_P^2 = 0.41$) as well as

SECPT-G performance ($F(1, 89) = 10.79$, $p = 0.001$, $\eta_p^2 = 0.11$) were found. Salivary α-amylase levels were highest between 21 and 21:30 pm and slightly decreased at later times (Fig. 3A). This was independent of the response to the SECPT-G. Participants who were classified as winners because they immersed their hands in the ice water for the maximally allowed time of 3 min, showed higher sAA levels at all three time points than participants who were classified as losers ($t_0$: $t(89) = 2.85$, $p = 0.005$, $d = -0.72$, $t_1$: $t(29.41) = 2.83$, $p = 0.008$, $d = -0.74$, $t_2$: $t(89) = 3.66$, $p = 0.164$, $d = -0.93$; Fig. 3B).

## DISCUSSION

Our study confirms that the SECPT-G is a well-suited experimental procedure for introducing an HPA-axis stress response. It, therefore, offers a very economical alternative to less economic stress induction set-ups like the TSST. However, in our study, no sAA response was found. Thus, when an sAA response is required, other set-ups (e.g., the TSST) might be better alternatives. The lack of sAA response in our study is unexpected, because a number of previous studies that investigated the effects of a cold-pressor test (CPT) without a socially-evaluative component did find sAA increases. In these studies, an sAA increase was found immediately after the CPT (*Skoluda et al., 2015*; *Van Stegeren, Wolf & Kindt, 2008*). A potential reason for our failure to find an sAA response might be that the study was performed in the late evening when a naturally decay in sAA levels takes place (*Nater et al., 2007*). This was also confirmed by the main effect of the factor time of day in our study. Furthermore, sAA levels are usually high (although they slightly decay) in the evening (*Nater et al., 2007*) and might have, therefore, masked or prevented an effect of our treatment. Future studies will, therefore, have to explore whether it is possible to induce sAA responses by means of the SECPT-G as performed in our experiment at other times of the day. Furthermore, it should be investigated whether the classical SECPT (not performed in groups) introduces an sAA response at different times of the day.

The cortisol response was independent of many anthropometric, demographic, and lifestyle factors as well as of time of day and immersion time as experimental factors. However, men showed a marginally different time course of the stress response than women. Basal cortisol levels at $t_2$ were by trend higher in men than in women, which corresponds to a pattern that is typically found (*Kirschbaum, Wüst & Hellhammer, 1992*; *Kudielka & Kirschbaum, 2005*; *Stephens et al., 2016*).

One further interesting finding is that participants who were classified as winners because they immersed their hand in the ice water for the maximally allowed time showed overall higher sAA levels than participants who were classified as losers. Although both groups showed no sAA increase in response to the SECPT-G, the lower levels in the losers group might be associated with lower overall arousal or with lower motivation which might have led to the worse performance during the SECPT-G. This should be further investigated in future studies.

Beside the late time of the day, our study is subject to some further limitations. One is that we did not use a control group which immersed their hands in warm water. Previous studies have shown—though with a slightly different procedure and with other

samples—that this does not introduce a stress response. Because our main goal was to show that the SECPT-G is a suitable application for studies in the general population and not that a warm water test introduces no response, this does not affect our conclusions much. However, there is a residual uncertainty that the stress response was not introduced by the SECPT-G itself, but by other situational factors (e.g., being in a laboratory for the first time, the test preparation phase or filling out the questionnaires) in our study.

Another limitation is that—although our sample is not the typical healthy student population at the age of early 20—it can be assumed that the people that came to our laboratory were interested in science and were, thus, still a specific population. Furthermore, the time point of the collection of the third saliva sample was quite early, in comparison to other studies that found the cortisol peak approximately 20 min after onset of the stressor (*Minkley et al., 2014*; *Schwabe, Haddad & Schachinger, 2008*). Therefore, it is very likely that cortisol levels would have increased further. However, since our study was conducted during a public event, it was not possible to investigate longer recovery periods. This will have to be done in future research. Moreover, our study design should be supplemented by collection of other stress markers (e.g., blood pressure, heart rate variability, inflammatory markers) in future research.

## CONCLUSIONS

Our study confirms that the SECPT-G is a stress induction tool which elicits a strong HPA axis response, which is mostly independent of many anthropometric, demographic, lifestyle, and experimental factors, and which can, therefore, be used for research in the general population. We conclude that the SECPT-G is particularly useful for studying the general population regardless of common exclusion factors which makes it a good means for clinical applications. In future research, it should be investigated whether the SECPT-G introduces an sAA response at earlier times of the day. Furthermore, other physiological stress markers (e.g., heart rate variability and inflammatory markers) should be included in future studies.

## ACKNOWLEDGEMENTS

We thank Aylin Gögsen and Elisa Merkenschlager for supporting data collection.

### Funding

Linda Becker was supported by the Bavarian Equal Opportunities Sponsorship—Förderung von Frauen in Forschung und Lehre (FFL)—Promoting Equal Opportunities for Women in Research and Teaching. We received support from the Deutsche Forschungsgemeinschaft and Friedrich-Alexander-Universität Erlangen-Nürnberg (FAU) within the funding programme Open Access Publishing. The funders had no role in study design, data collection and analysis, decision to publish, or preparation of the manuscript.

## Grant Disclosures

The following grant information was disclosed by the authors:
Bavarian Equal Opportunities Sponsorship—Förderung von Frauen in Forschung und Lehre (FFL)—Promoting Equal Opportunities for Women in Research and Teaching. Deutsche Forschungsgemeinschaft and Friedrich-Alexander-Universität Erlangen-Nürnberg (FAU).

## Competing Interests

The authors declare that they have no competing interests.

## Author Contributions

- Linda Becker conceived and designed the experiments, performed the experiments, analyzed the data, contributed reagents/materials/analysis tools, prepared figures and/or tables, authored or reviewed drafts of the paper, approved the final draft.
- Ursula Schade conceived and designed the experiments, performed the experiments, contributed reagents/materials/analysis tools, approved the final draft.
- Nicolas Rohleder contributed reagents/materials/analysis tools, authored or reviewed drafts of the paper, approved the final draft.

## Human Ethics

The following information was supplied relating to ethical approvals (i.e., approving body and any reference numbers):

The study was approved by the local ethics committee of the Friedrich-Alexander University Erlangen-Nuremberg (6_18 B).

## Data Availability

The data is available in the Supplemental File.

## Supplemental Information

Supplemental information for this article can be found online at http://dx.doi.org/10.7717/peerj.7521#supplemental-information.

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
