# Peer review of "Evaluation of the socially evaluated cold-pressor group test (SECPT-G) in the general population"

_PeerJ, doi:10.7717/peerj.7521_

## Round 0.1 · original submission · Major Revisions

I now have received three reviewers' comments. Although all reviewer expressed their interest in your study, several aspects of this manuscript should be revised to improve its clarity. Their observations are presented with clarity so I'll not risk confusing matters by belaboring or reiterating their comments. While I might quibble with the occasional point, I note that I regard the reviewers' opinions as substantive and well-informed. I believe that all of the highlighted reservations require contemplation and appropriate attention in revising the document if it is to contribute appropriately to Peerj and the extant literature. Please revise or refute according to the two reviewers' comments and provide a point by point reply in addition to the revised manuscript.


Tsung-Min Hung, PhD., FNAK, FISSP
PeerJ editor
Research chair professor,
Department of Physical Education,
National Taiwan Normal University

Reviewer 1 ·

Basic reporting

The study presented here aimed to determine the effects of the socially evaluated cold-pressor test (SECPT) on acute stress responses (subjective stress perception, saliva cortisol, salivary alpha-amylase) in groups. Performing this test in small groups of the "broad population", including participants of different sex, age, with different socioeconomic background, etc., and correcting measurements of stress responses for these and other covariates, would be highly economic relevant by enabling the determination of acute stress responses in a high number of people at once.

In general, the manuscript is very well written and in very good English.
The authors cite a high number of relevant literature, provide all relevant background information in the introduction, and relate their results to previous findings without being too speculative at any stage.
The raw data is supplied as supplementary file.

There are just a few points (see general comments), which might help improving the introduction and discussion.

Experimental design

The manuscript presents primary research. The research questions are well defined, relevant and meaningful. I think the experiment was in general well designed and also shows that the SECPT can be performed in a very short time frame and in a high number of participants, which definitely proved the validity of the SECPT-G.
The study has been approved by the responsible ethics committee.

The methods are well described, there a just a few minor concerns that are adressed below (see general comments).

However, I have some major concerns regarding the analyses (see validity of the findings below).

Validity of the findings

Although the results definitely confirm the biological/psychological validitiy of the SECPT-G, there are some major concerns regarding the analyses and the results:

One thing that confuses me overall is that the variances (standard error, standard deviation, confidence interval ?) for all parameters presented in figure 1 are seemingly not the same as presented in the text. The means presented in the text and in the figures are obviously the same, but the variances presented in the text are much higher than in the figures. Please either explain in the figure captions, which values are shown, or at least in the statistics section of the methods, which variances you provide in the figures and in the text.
Similarly, also the Eta-squares values for the cortisol and sAA analyses do not fully match when comparing the values presented in the text and in table 1; see for example eta-square for cortisol analysis in the text (= .23; line 194/195) and in table 1 (= .37). Do these values result from different analyses? If the respective values should be the same then I would recommend presenting them either in the text or the figures/table. This also applies to the further analyses, when covariates were included.

How many rmANOVAs were calculated? Based on your "Statistical analysis" section in the Methods as well as different p- and eta-square values for the main effect of time in table 1, I assume that new rmANOVAs on cortisol and sAA measurements were performed for each single covariate, so in total 11 additional rmANOVAs for cortisol and 11 additional rmANOVAs for sAA, corresponding to the 11 covariates. If I am right, this would be a very unusual way of analysis. In order to detect any relevant effects of covariates, the usual analysis would be to include all covariates in the model at once, therefore correcting for all the other covariates (when calculating type 3 sum of squares), and perhaps removing non-relevant predictors/covariates from the model (for instance based on the Akaike information criterion). Some covariates may only show significance after correcting for others, or even do not show significance anymore. Including all possible covariates/predictors at once would be highly important to extract only the relevant ones, because they all apply to the participants at the same time and might even be related to each other (e.g. physical activity and BMI). Did you control for this? Please clarify, how the data were analyzed.

Furthermore, were all covariates include as numerical/continuous variables (e.g. 0,1,2,3,...)? In case of graduation or profession, these variables definitely have to be included as categorial predictors ("groups") in the analyses, because it cannot be assumed that the measurements of stress responses change linearly with the numbering, but differences between these "groups" (e.g. student, employee, retired, other) have to be considered in any way. Please also clarify this.

Additionally, in line 139 it is said that it was "(...) the aim to introduce a competitive situation". Could it be possible that some kind of "winner-loser-effect" affected your measurements of stress responses? Perhaps, a participant who pulled his/her hand out of the water earliest in the group (the "loser") would show a different stress response (presumably a change in cortisol concentrations) compared to someone who had an immersion time of 3 minutes in the same group (the "winner"). In this case not only the own immersion time of a participant might affect his/her stress response, but also the own immersion time in relation to the immersion times of the other participants in the group. Is it possible/meaningful to include this in the analyses too?

Additional comments

Minor concerns/comments

Introduction

The provided information for effects of demographic, socioeconomic and lifestyle factors on stress responses in lines 54-70 is somehow just a "listing". I think this section should be re-written and/or shortened and the findings/information should be set in relation to each other.

Methods

In the "Participants" section of the Methods you provide relevant information, but how most of these data were recorded is stated afterwards in the "Procedure" section. Perhaps, it would be better to describe the general procedure first.
Line 173: What do you mean with "If necessary, Cohen's d was corrected (...)" ?

Disussion

In the "Participants" section of the Methods you provide relevant information, but how most of these data were recorded is stated afterwards in the "Procedure" section. Perhaps, it would be better to describe the general procedure first so that it clear from the beginning, where these data come from (e.g. measurements, questionnaires, etc.).

Line 173: What do you mean with "If necessary, Cohen's d was corrected (...)" ?

Reviewer 2 ·

Basic reporting

In general the structure is good. Some minor changes:
- line 20 - misses (SAA)
- line 33 please explain better why provoking stress responses in subjects is an important tool in bio-behavoral research
- line 47 - please explain what "strong" means; this is not a good word; what happen to this physiological markers during stress (e.g. HRV decreases; BP increases...) ?
- line 94 - The correct way is: the sample size is 96 (N=96); the same for line 97.

The English needs to be reviewed by a professional.

The literature references are outdated.

Experimental design

The methodology is well described

Just one question
Did you conduct a normality test before choosing the parametric test ANOVA?

Validity of the findings

Data is robust, and conclusions are well stated.

Additional comments

Very interesting paper for stress research area. The sample is robust and the results are well presented and discussed.

Reviewer 3 ·

Basic reporting

The manuscript is clearly written and easy to follow.

The study rationale is clearly addressed.

Experimental design

1. The research question is well defined.

2. The experimental procedures should be clarified.
2.1. Methods described is unclear. Please describe the assessments of demographic variables and lifestyle factors in greater detail. Also, please provide validity and reliability data of the TICS-SSCS and IPAQ-short form. In a related note, how you 1) defined smokers and non-smokers? 2) how BMI was assessed? by self-reported or following standard laboratory procedure?
2.2. The reporting of activity levels is problematic. When using the IPAQ as measure of activity levels, we usually report the results using metabolic equivalents (METs) per week, not metabolic minutes per week. Please clarify this.

3. The current study did not include a control group. As noted by the authors themselves, there were other factors that could have also introduced stress response, the authors are encouraged to have reflection on this issue.

Validity of the findings

The reviewer has several concerns about the validity of findings.

1. Did the authors perform a priori power calculation to determine the sample size?

2. Do BMI norms change by age? If so, how the authors deal with this matter?

3. Participants were asked to refrain from alcohol for only 2 hours before experiment or to consume no more than two drinks (which is pretty vague). As the half-life of alcohol could be longer than 2 hours, depends on the amount of consumption, the 2-hour restriction is not rigorous enough. In a related note, participants were not refrained from caffeine before experiment, this is also problematic as caffeine is a SNS stimulant.

4. The authors reported ethnic data but did not make use of them. It might be interesting to look at whether ethnic affects the results.

5. Participants were asked to fill out several questionnaires between t1 and t2 measure points. Is it possible that filling out these inventories affects following sAA/cortisol measures in t2?

---

## Round 0.2 · accepted · Accept

I have now received all the reviewers’ comment with satisfaction of your reply and revisions from previous comments. You and your coauthors have my congratulations. Thank you for choosing PeerJ as a venue for publishing your research work and I look forward to receiving more of your work in the future.

Tsung-Min Hung, PhD., FNAK, FISSP
PeerJ editor
Research chair professor,
Department of Physical Education,
National Taiwan Normal University

Reviewer 1 ·

Basic reporting

The manuscript is very well written and the English was further improved.
Additional literature was included to provide even more background.
All tables and figures are clear, raw data is provided.
The conducted changes definitely improved the manuscript's quality.

Experimental design

The manuscript presents primary research, the research questions are well defined, relevant and meaningful.
The experiment was well designed, the conducted changes to the methods section further improved the clarity on how specific procedures, measurements, and analyses were performed.
The study has been approved by the responsible ethics committee. All relevant information with regard to ethical issues are provided.

Validity of the findings

The authors performed a variety of additional analyses (additional RM-ANOVAs, Power analyses, etc.) in order to confirm that the SECPT-G can be performed in a general population based on significant HPA-axis/cortisol responses.
The findings and conclusions are definitely valid and are linked to original research questions.

The authors responded to all raised concerns adequately and answered all my questions.

Reviewer 3 ·

Basic reporting

no comment

Experimental design

no comment

Validity of the findings

no comment

Additional comments

The authors have addressed all my concerns. I am now satisfied with the manuscript.